A nomogram for predicting bladder dysfunction in patients with type 2 diabetes mellitus: a retrospective study

Hu Yingjie 1 2
http://orcid.org/0009-0005-9989-268X Hao Fengming 3
Wang Ying 1 2
Chen Ling 1 2
Wen Lihua 4
Li Jue 5
Ren Wei 1 2
Cai Wenzhi 1 2 Caiwwenz@163.com
1 Department of Nursing, Shenzhen Hospital, Southern Medical University , Shenzhen, Guangdong , China
2 School of Nursing, Southern Medical University , Guangzhou, Guangdong , China
3 School of Nursing, Shanxi Technology and Business University , Shanxi, Taiyuan , China
4 Department of Hepatobiliary, Suzhou Ninth People’s Hospital , Suzhou , China
5 SUMC Center for Nursing Research, Shantou University Medical College , Shantou , China
Foti Daniela
Electronic publication date: 2025 Jan 22
Publication date: 2025
Volume: 13
Electronic Location ID: e18872
Received 2024 Sep 13; Accepted 2024 Dec 26
Copyright: © 2025 Hu et al.
Copyright year: 2025
Copyright holder: Hu et al.
License: This is an open access article distributed under the terms of the Creative Commons Attribution License, which permits unrestricted use, distribution, reproduction and adaptation in any medium and for any purpose provided that it is properly attributed. For attribution, the original author(s), title, publication source (PeerJ) and either DOI or URL of the article must be cited.
License URL: https://creativecommons.org/licenses/by/4.0/

Keywords: Type 2 diabetes mellitus, Diabetic bladder dysfunction, Factor, Predictive model

Funding: Sanming Project of Medicine in Shenzhen, China SZSM202211044 Shenzhen Science and Technology Project JCYJ 20210324142406016 SUMC Scientific Research Foundation for Talents 510858069 Ninth People’s Hospital of Suzhou YK202236 This work was supported by the Sanming project of medicine in Shenzhen, China (Grant Number SZSM202211044); the Shenzhen Science and Technology Project (Grant Number JCYJ 20210324142406016); the SUMC Scientific Research Foundation for Talents (Grant Number 510858069), and project of scientific research initiation fund of the Ninth People’s Hospital of Suzhou (Grant Number YK202236). The funders had no role in study design, data collection and analysis, decision to publish, or preparation of the manuscript.

==============================
Background

Diabetic bladder dysfunction (DBD) is a common urinary complication in diabetic patients, significantly affecting their overall well-being and quality of life, and placing a considerable burden on healthcare resources. Early prevention is crucial; however, the absence of a simple and effective tool to predict DBD onset remains a significant challenge. This study aims to identify risk factors for DBD in patients with type 2 diabetes mellitus (T2DM) and to develop a predictive nomogram for clinical application.

Methods

This retrospective study included patients with T2DM treated at two hospitals. Data from patients treated at one hospital between January 2020 and August 2023 were used to create the training set, while data from patients treated at another hospital between March 2022 and October 2023 were used to create the validation set. Patients were classified into two groups based on the presence or absence of DBD: the DBD group and the non-DBD group. Significant factors identified via bivariate analysis (P < 0.05) were incorporated into multivariate logistic regression to construct a predictive model, and a corresponding nomogram was developed. The model’s performance was assessed using receiver operating characteristic (ROC) curves, calibration plots, decision curve analysis (DCA), and clinical impact plots (CIC), with validation performed through 1,000 bootstrap resamplings.

Results

A total of 1,010 participants were included in this study, with a DBD incidence rate of 38.81% (392/1,010). Multivariate logistic regression analysis identified HbA1c, PCP-2h, DPN, TCO2, PAB, T-Bil, I-Bil, IgE, URBC, UI and UR as independent risk factors for DBD. A nomogram was constructed based on these factors. Both internal and external validations demonstrated the good predictive performance of the nomogram. The area under the curve (AUC) for the training and validation datasets was 0.897 and 0.862, respectively. The calibration curve showed a high degree of consistency. Results from DCA and CIC indicated that the prediction model had high clinical utility.

Conclusions

A predictive model and nomogram for DBD in T2DM patients were developed, demonstrating strong accuracy and clinical utility, aiding in early DBD risk assessment and intervention.

Introduction

Type 2 diabetes mellitus (T2DM) is among the most prevalent chronic diseases globally, affecting approximately 529 million individuals as of 2021. By 2050, this number is projected to exceed 1.31 billion (GBD 2021 Diabetes Collaborators, 2023). Diabetic bladder dysfunction (DBD) is a common and serious complication of T2DM that significantly impacts patients’ physical and mental health (Xu et al., 2017). Studies indicate that 25–87% of T2DM patients experience varying degrees of DBD (Wittig et al., 2019). Also referred to as diabetic neurogenic bladder (Moller, 1976), DBD is characterized by impaired bladder sensory nerves, reduced detrusor muscle contractility, increased bladder capacity, and elevated residual urine volume (Wittig et al., 2019). This condition can lead to upper urinary tract damage, including pyelonephritis, hydronephrosis, and ureteral dilation (Nseyo & Santiago-Lastra, 2017). Unfortunately, once diagnosed, DBD is typically irreversible, and treatment outcomes are often unsatisfactory (Bree & Santiago-Lastra, 2020). Therefore, early intervention is crucial for managing DBD and improving patient outcomes.

Currently, urodynamic tests, post-void residual urine (PVR) measurement, and symptom assessment scales are the primary methods for diagnosing and evaluating DBD. Guidelines recommend these three methods as tools for assessing DBD risk (Guo et al., 2020; Ginsberg et al., 2021). However, several challenges exist in clinical practice. Both urodynamic tests and PVR measurement rely on specialized medical equipment, and urodynamic tests are invasive procedures, limiting their widespread use, particularly in primary healthcare settings. Additionally, while symptom assessment scales are easy to use, their objectivity and accuracy are significantly influenced by subjective factors, posing further limitations. Therefore, there is an urgent need to develop a convenient, objective, and user-friendly tool for predicting DBD risk.

However, there are limited predictive tools for assessing the risk of DBD in T2DM patients. In response to this gap, the present study comprehensively and systematically collected relevant data from T2DM patients using a retrospective study design. For the first time, general clinical data, renal function, hepatic function, lipid profiles, immune function, and urine and blood indices were all included in the analysis. This approach not only broadened the scope of information but also enhanced the accuracy of the predictive model. The findings aim to provide an effective tool for identifying high-risk groups for DBD, thereby aiding in the prevention and reduction of DBD incidence.

Materials and Methods

Study population

This retrospective study was conducted at two comprehensive hospitals in Shenzhen, China, utilizing electronic medical records from Southern Medical University Shenzhen Hospital between January 2020 and August 2023 to identify T2DM cases. These records formed the training set, while the validation set was composed of data from patients treated at Shenzhen Third People’s Hospital from March 2022 to October 2023.

Inclusion criteria: (1) Diagnosed with T2DM (ICD-10 code E11.900); (2) comprehensive medical records documenting both bladder symptoms and T2DM-related manifestations.

Exclusion criteria: (1) Duration of diabetes <5 years; (2) neuropathy due to non-diabetic causes (e.g., spinal cord injury, multiple sclerosis (MS), stroke, spina bifida, Parkinson’s disease); (3) urinary tract infection (UTI) within the past month; (4) prostate-related conditions (e.g., benign prostatic hyperplasia (BPH), prostate cancer, history of prostate surgery); (5) pelvic conditions (e.g., pelvic organ prolapse, history of pelvic surgery); (6) acute metabolic complications of diabetes (e.g., diabetic ketoacidosis, hyperosmolar hyperglycemic state); (7) severe dysfunction of the heart, liver, lungs, or kidneys. Only data from the initial admission were included. The exclusion process is illustrated in Fig. 1.

Figure 1 Flowchart of exclusion process.

Note: (A) represents the flowchart of the training set; (B) represents the flowchart of the validation set. Abbreviations: T2DM, Type 2 diabetes mellitus; DBD, Diabetic bladder dysfunction.

Sampling

Referring to the rough estimation method for sample size in multiple logistic regression with multiple factors: for the class with a lower proportion in the outcome variable, the sample size should be at least 10 times the number of independent variables plus a constant factor (Rao, 2003). In this study, the dependent variable has two levels (DBD and Non-DBD), and a preliminary estimate suggests there are 16 significant independent variables. Therefore, the sample size for the case group in this study is approximately 16 × 10 = 160 cases. Based on the literature, the incidence rate of DBD is reported to range between 25% and 87% (Wittig et al., 2019; Moussa et al., 2020). Using 25% as the estimated incidence rate for DBD in this study, the required sample size is calculated as 160 ÷ 25% = 640 cases. The training and validation sets were divided in a 7:3 ratio. With the training set requiring at least 640 cases, the validation set must contain at least 275 cases.

Study variables

Outcome

The diagnosis of DBD was based on the following criteria: (1) Confirmation of a T2DM diagnosis; (2) presence of one or more lower urinary tract symptoms, such as urinary frequency, urgency, polyuria, increased nocturia, dysuria, incontinence, or urinary retention; (3) a temporal correlation between T2DM and a bladder residual urine volume ≥50 mL, as determined by B-ultrasound (Guo et al., 2020); (4) DBD symptoms that could not be attributed to other causes.

Based on the presence or absence of DBD, the subjects were categorized into two groups: the case group (DBD) and the control group (Non-DBD). Clinical data from both groups were then compared for analysis.

Potential predictive factor

(1) Demographic information

In this section, we overview collected demographic data, including age, gender, occupation, marital status, education, insurance type, and BMI.

(2) Clinical and bladder symptomatic information

This section focuses on the collection of essential clinical and bladder symptomatic information related to T2DM. The data collected include the duration of T2DM, the use of duration of T2DM, oral hypoglycemic agents, insulin injections, mecobalamin supplementation, and key clinical indicators such as glycated hemoglobin (HbA1c), fasting blood glucose (FBG), postprandial blood glucose at 2 hours (PBG-2h), postprandial insulin at 2 hours (PPI-2h), and postprandial c-peptide at 2 hours (PCP-2h) were measured. Furthermore, complications such as diabetic peripheral neuropathy (DPN), diabetic retinopathy, diabetic nephropathy (DN), hypertension, and coronary heart disease were recorded. Bladder symptoms including urinary frequency, urgency, polyuria, nocturia, dysuria, urinary incontinence (UI), and urinary retention (UR) were also documented.

(3) Laboratory examination

The laboratory examination primarily consists of six categories: renal function, liver function, blood lipid levels, immune function, urinalysis, and blood routine indicators.

Renal function indicators include urine microalbumin/creatinine ratio (UA/CR), 24-hour urine microalbumin (UMA-24h), 24-hour urine volume (UV-24h), 24-hour urine protein quantification (UPQ-24h), serum urea (SU), serum creatinine (SC), serum uric acid (SUA), total carbon dioxide (TCO2), glomerular filtration rate (GFR), cystatin C (CysC), and β2-microglobulin (β2-MG).

Liver function indicators include total protein (TP), albumin (ALB), prealbumin (PAB), total bilirubin (T-BIL), direct bilirubin (D-BIL), and indirect bilirubin (I-BIL).

Blood lipid levels includes triglycerides (TG), total cholesterol (TC), high-density lipoprotein cholesterol (HDL-C), and low-density lipoprotein cholesterol (LDL-C).

Immune function indicators include high-sensitivity C-reactive protein (HS-CRP), IgG, IgA, IgM, IgE, complement C3 (C3), and complement C4 (C4).

Urinalysis includes specific gravity (SG), pH, nitrite (Nit), protein (Pro), glucose (Glu), ketone bodies (Ket), urobilinogen (UBG), occult blood (OB), urine red blood cells (URBC), and urine white blood cells (UWBC).

Blood indicators includes red blood cell count (RBC), white blood cell count (WBC), platelets (PLT), absolute neutrophil count (ANC), neutrophil percentage (NEUT%), absolute lymphocyte count (ALC), lymphocyte percentage (Lymph%), monocyte percentage (MONO%), absolute monocyte count (AMC), and erythrocyte sedimentation rate (ESR).

Data collection

Data collection was conducted by professional researchers, who strictly adhered to inclusion and exclusion criteria when extracting patient medical records. They did not participate in the subsequent statistical analysis. To avoid duplication of data from the same patient, only the first hospitalization of each patient with multiple admissions due to T2DM was included. Additionally, the training set consisted of electronic medical records of T2DM patients treated at Southern Medical University Shenzhen Hospital from January 2020 to August 2023, while the validation set included data from patients treated at Shenzhen Third People’s Hospital from March 2022 to October 2023.

Statistical analysis

We conducted statistical analyses using SPSS 26.0 software and R (Version 4.3.1), with a significance level set at α = 0.05. Continuous data adhering to a normal distribution were presented as mean ± standard deviation, and group differences were assessed using the two-independent-sample t-test. Non-normally distributed continuous data were presented as median (quartiles), and group differences were compared using the Mann-Whitney U test. Categorical data were presented as frequencies, and percentages (%), with group differences analyzed using the χ2 test.

Variables showing significant differences in the univariate analysis (P < 0.05) were further analyzed using a multifactorial logistic regression analysis to identify the risk factors and establish the prediction model. Nomograms were constructed using the ‘rms’ package in R software (Version 4.4.0; R Core Team, 2024). The discriminative ability and calibration of the model were assessed through receiver operating characteristic (ROC) curves, Hosmer-Lemeshow (H-L) goodness-of-fit tests, and calibration curves. Internal validation used 1,000 bootstrap resamplings. Additionally, decision curve analysis (DCA) and clinical impact curve (CIC) were employed to evaluate the clinical utility of the model.

Ethical considerations

Medical records were stored on dedicated hard drives, accessible only to authorized team members. Patient identification information was removed to ensure privacy. Researchers were prohibited from conducting statistical analyses online or transmitting data over the internet. This study was approved by the Ethics Committee of Shenzhen Hospital of Southern Medical University (No.: NYSZYYEC20230086) and informed consent was waived.

Results

Study participants

After applying the inclusion and exclusion criteria, the final data analysis included 1,010 participants, with a median age of 57.00 years. Among them, 392 patients developed DBD, with an incidence rate of 38.81% (392/1,010). The training set consisted of 723 patients, of whom 285 (39.42%) developed DBD. The validation set included 287 patients, with 107 (37.28%) developing DBD. Detailed patient characteristics are presented in Table 1.

Table 1 Comparison of relevant factors in patients with T2DM.

Variables	Training (n = 723)	Validation (n = 287)	Total
(n = 1,010)	Statistic	P-
Value	
Non-DBD
(n = 438)	DBD
(n = 285)	Statistic	P-
Value	Non-DBD
(n = 180)	DBD
(n = 107)	Statistic	P-Value	
Demographic information											
Age, M (Q1, Q3), years	56.00
(49.00, 64.00)	59.00 (51.00, 68.00)	Z = −2.81	0.005*	56.50
(48.00, 65.00)	57.00
(49.50, 68.50)	Z = −0.55	0.585	57.00
(49.00, 66.00)	Z = −0.66	0.509	
Gender, n (%)			χ2 = 2.61	0.106			χ2 = 1.11	0.293		χ2 = 0.03	0.860	
Male	282 (64.38)	200 (70.18)			117 (65.00)	76 (71.03)			675 (66.83)			
Female	156 (35.62)	85 (29.82)			63 (35.00)	31 (28.97)			335 (33.17)			
Occupation, n (%)			χ2 = 0.23	0.890			χ2 = 5.83	0.054		χ2 = 1.63	0.443	
Employed	37 (8.45)	24 (8.42)			11 (6.11)	11 (10.28)			83 (8.22)			
Unemployed	59 (13.47)	42 (14.74)			25 (13.89)	24 (22.43)			150 (14.85)			
Other	342 (78.08)	219 (76.84)			144 (80.00)	72 (67.29)			777 (76.93)			
Marital status, n (%)			χ2 = 0.47	0.925			–	1.000		χ2 = 3.59	0.309	
Unmarried	10 (2.28)	7 (2.46)			3 (1.67)	2 (1.87)			22 (2.18)			
Married	409 (93.38)	265 (92.98)			166 (92.22)	99 (92.52)			939 (92.97)			
Divorced	8 (1.83)	4 (1.40)			6 (3.33)	4 (3.74)			22 (2.18)			
Widowed	11 (2.51)	9 (3.16)			5 (2.78)	2 (1.87)			27 (2.67)			
Education level, n (%)			χ2 = 1.78	0.775			χ2 = 1.59	0.810		χ2 = 7.81	0.099	
Illiterate	40 (9.13)	27 (9.47)			10 (5.56)	6 (5.61)			83 (8.22)			
Primary school	100 (22.83)	69 (24.21)			45 (25.00)	26 (24.30)			240 (23.76)			
Middle school	111 (25.34)	81 (28.42)			42 (23.33)	29 (27.10)			263 (26.04)			
Senior high school	142 (32.42)	83 (29.12)			54 (30.00)	34 (31.78)			313 (30.99)			
University and above	45 (10.27)	25 (8.77)			29 (16.11)	12 (11.21)			111 (10.99)			
Insurance type, n (%)			χ2 = 1.12	0.290			χ2 = 0.00	0.991		χ2 = 0.10	0.756	
Self-Pay	64 (14.61)	50 (17.54)			27 (15.00)	16 (14.95)			157 (15.54)			
Insured	374 (85.39)	235 (82.46)			153 (85.00)	91 (85.05)			853 (84.46)			
BMI (kg/cm2), M (Q1, Q3)	23.58
(21.78, 26.22)	22.99
(21.34, 25.78)	Z = −1.87	0.062	24.01
(21.94, 26.56)	23.80
(21.42, 26.45)	Z = −0.47	0.637	23.63
(21.62, 26.17)	Z = −1.87	0.061	
Note:

Categorical data are shown as frequencies (%) and continuous data as median (quartile); Z: Mann-Whitney test; χ2: Chi-square test; −: Fisher exact; the asterisk (*) indicates that the result is statistically significant (P < 0.05).

Comparison of relevant factors in patients with T2DM

This study examined 76 variables, with demographic information provided in Table 1, clinical and bladder symptom data comprehensively summarized in Table 2, and laboratory examination results systematically presented in Table 3. In the training set, bivariate analysis identified 23 variables significantly associated with the occurrence of DBD in patients with T2DM. These variables included age, HbA1c, PPI-2h, PCP-2h, DPN, coronary heart disease, urinary frequency, polyuria, UI, UR, UMA-24h, TCO2, PAB, T-Bil, I-Bil, IgG, IgA, IgE, C4, SG, URBC, UWBC, and mono%.

Table 2 Clinical and bladder symptomatic information in patients with T2DM.

Variables	Training (n = 723)	Validation (n = 287)	Total
(n = 1,010)	Statistic	P-
Value	
Non-DBD
(n = 438)	DBD
(n = 285)	Statistic	P-
Value	Non-DBD
(n = 180)	DBD
(n = 107)	Statistic	P-Value	
Management indicators										
Duration of T2DM (years),
M (Q1, Q3)	10.00
(6.50, 15.00)	10.00
(7.00, 15.00)	Z = −0.53	0.598	8.00
(4.00, 12.25)	10.00
(3.00, 12.00)	Z = −0.32	0.749	10.00
(6.00, 14.00)	Z = −4.68	<0.001*	
Oral hypoglycemic agents,
n (%)			χ2 = 2.48	0.116			χ2 = 9.90	0.002*		χ2 = 0.22	0.643	
No	186 (42.47)	138 (48.42)			65 (36.11)	59 (55.14)			448 (44.36)			
Yes	252 (57.53)	147 (51.58)			115 (63.89)	48 (44.86)			562 (55.64)			
Insulin injection, n (%)			χ2 = 1.87	0.171			χ2 = 0.15	0.697		χ2 = 1.71	0.190	
No	49 (11.19)	23 (8.07)			14 (7.78)	7 (6.54)			93 (9.21)			
Yes	389 (88.81)	262 (91.93)			166 (92.22)	100 (93.46)			917 (90.79)			
Mecobalamin, n (%)			χ2 = 0.00	0.961			χ2 = 2.01	0.156		χ2 = 0.39	0.532	
No	262 (59.82)	171 (60.00)			106 (58.89)	72 (67.29)			611 (60.50)			
Yes	176 (40.18)	114 (40.00)			74 (41.11)	35 (32.71)			399 (39.50)			
HbA1c, M (Q1, Q3), (%)	8.60
(7.20, 10.70)	9.20
(7.50, 12.00)	χ2 = −3.16	0.002*	8.50
(7.47, 10.40)	9.70
(7.85, 12.25)	Z = −3.40	<0.001*	8.90
(7.30, 11.20)	Z = −0.34	0.731	
FBG, M (Q1, Q3), (mmol/L)	6.53
(5.29, 8.05)	6.88
(5.44, 8.26)	Z = −1.94	0.053	6.57
(5.38, 8.16)	7.00
(5.44, 9.16)	Z = −1.51	0.132	6.68
(5.39, 8.27)	Z = −0.98	0.326	
PBG-2h, M (Q1, Q3), (mmol/L)	15.65
(12.14, 18.90)	15.15
(13.01, 18.31)	Z = −0.51	0.608	15.16
(12.02, 18.90)	15.35
(12.22, 17.52)	Z = −0.97	0.334	15.39
(12.40, 18.43)	Z = −0.48	0.634	
PPI-2h, M (Q1, Q3), (μIU/mL)	78.05
(29.01, 245.25)	52.32
(22.58, 147.00)	Z = −3.37	0.001*	84.00
(31.53, 258.00)	53.21
(22.86, 170.00)	Z = −2.28	0.023*	68.38
(25.48, 221.00)	Z = −0.59	0.555	
PCP-2h, M (Q1, Q3), (ng/ml)	2.50
(1.44, 4.43)	2.06
(0.93, 3.73)	Z = −2.84	0.005*	2.59
(1.41, 4.13)	2.17
(1.03, 3.70)	Z = −1.55	0.121	2.35
(1.20, 4.11)	Z = −0.17	0.866	
Complications												
DPN, n (%)			χ2 = 60.91	<0.001*			χ2 = 25.67	<0.001*		χ2 = 9.23	0.002*	
No	195 (44.52)	47 (16.49)			25 (13.89)	43 (40.19)			310 (30.69)			
Yes	243 (55.48)	238 (83.51)			155 (86.11)	64 (59.81)			700 (69.31)			
Diabetic retinopathy, n (%)			χ2 = 0.56	0.456			χ2 = 1.32	0.251		χ2 = 0.18	0.674	
No	287 (65.53)	179 (62.81)			123 (68.33)	66 (61.68)			655 (64.85)			
Yes	151 (34.47)	106 (37.19)			57 (31.67)	41 (38.32)			355 (35.15)			
Diabetic nephropathy, n (%)			χ2 = 0.07	0.789			χ2 = 0.04	0.850		χ2 = 0.22	0.637	
No	339 (77.40)	223 (78.25)			143 (79.44)	84 (78.50)			789 (78.12)			
Yes	99 (22.60)	62 (21.75)			37 (20.56)	23 (21.50)			221 (21.88)			
Hypertension, n (%)			χ2 = 0.02	0.875			χ2 = 0.11	0.745		χ2 = 1.70	0.192	
No	227 (51.83)	146 (51.23)			86 (47.78)	49 (45.79)			508 (50.30)			
Yes	211 (48.17)	139 (48.77)			94 (52.22)	58 (54.21)			502 (49.70)			
Coronary heart disease, n (%)			χ2 = 3.87	0.049*			χ2 = 2.25	0.133		χ2 = 0.41	0.524	
No	364 (83.11)	252 (88.42)			152 (84.44)	97 (90.65)			865 (85.64)			
Yes	74 (16.89)	33 (11.58)			28 (15.56)	10 (9.35)			145 (14.36)			
Bladder symptoms												
Urinary frequency, n (%)			χ2 = 6.56	0.010*			χ2 = 1.90	0.168		χ2 = 1.18	0.278	
No	280 (63.93)	155 (54.39)			96 (53.33)	66 (61.68)			597 (59.11)			
Yes	158 (36.07)	130 (45.61)			84 (46.67)	41 (38.32)			413 (40.89)			
Urinary urgency, n (%)			χ2 = 0.00	0.993			χ2 = 0.59	0.442		χ2 = 4.37	0.037*	
No	286 (65.30)	186 (65.26)			127 (70.56)	80 (74.77)			679 (67.23)			
Yes	152 (34.70)	99 (34.74)			53 (29.44)	27 (25.23)			331 (32.77)			
Polyuria, n (%)			χ2 = 12.20	<0.001*			χ2 = 0.37	0.544		χ2 = 4.83	0.028*	
No	286 (65.30)	149 (52.28)			124 (68.89)	70 (65.42)			629 (62.28)			
Yes	152 (34.70)	136 (47.72)			56 (31.11)	37 (34.58)			381 (37.72)			
Nocturia, n (%)			χ2 = 0.39	0.533			χ2 = 14.39	<0.001*		χ2 = 41.84	<0.001*	
No	279 (63.70)	175 (61.40)			88 (48.89)	28 (26.17)			570 (56.44)			
Yes	159 (36.30)	110 (38.60)			92 (51.11)	79 (73.83)			440 (43.56)			
Dysuria, n (%)			χ2 = 3.71	0.054			χ2 = 7.18	0.007*		χ2 = 3.81	0.051	
No	373 (85.16)	227 (79.65)			149 (82.78)	74 (69.16)			823 (81.49)			
Yes	65 (14.84)	58 (20.35)			31 (17.22)	33 (30.84)			187 (18.51)			
UI, n (%)			χ2 = 12.75	<0.001*			χ2 = 6.45	0.011*		χ2 = 5.70	0.017*	
No	293 (66.89)	153 (53.68)			135 (75.00)	65 (60.75)			646 (63.96)			
Yes	145 (33.11)	132 (46.32)			45 (25.00)	42 (39.25)			364 (36.04)			
UR, n (%)			χ2 = 78.29	<0.001*			χ2 = 0.04	0.845		χ2 = 3.40	0.065	
No	349 (79.68)	137 (48.07)			131 (72.78)	79 (73.83)			696 (68.91)			
Yes	89 (20.32)	148 (51.93)			49 (27.22)	28 (26.17)			314 (31.09)			
Notes:

Categorical data are shown as frequencies (%) and continuous data as median (quartile); Z: Mann-Whitney test; χ2: Chi-square test.

HbA1c, Glycated Hemoglobin; FBG, Fasting Blood Glucose; PBG-2h, Postprandial Blood Glucose 2-hour; PPI-2h, Postprandial Insulin 2-hour; PCP-2h, Postprandial C-peptide 2-hour; DPN, Diabetic Peripheral Neuropathy; UI, Urinary Incontinence; UR, Urinary Retention; the asterisk (*) indicates that the result is statistically significant (P < 0.05).

Table 3 Laboratory examination information of patients with T2DM.

Variables	Training (n = 723)	Validation (n = 287)	Total
(n = 1,010)	Statistic	P-
Value	
Non-DBD
(n = 438)	DBD
(n = 285)	Statistic	P-
Value	Non-DBD
(n = 180)	DBD
(n = 107)	Statistic	P-Value	
Renal function indicators											
UA/CR, n (%), (mg/g)			χ2 = 0.46	0.794			χ2 = 9.44	0.009*		χ2 = 3.93	0.140	
Normal	357 (81.51)	229 (80.35)			162 (90.00)	82 (76.64)			830 (82.18)			
Microalbuminuria	72 (16.44)	48 (16.84)			14 (7.78)	20 (18.69)			154 (15.25)			
Macroalbuminuria	9 (2.05)	8 (2.81)			4 (2.22)	5 (4.67)			26 (2.57)			
UMA-24, M (Q1, Q3), (mg/g)	18.04
(6.11, 100.16)	28.35
(10.24, 102.90)	Z = −2.80	0.005*	17.99
(5.82, 64.92)	28.98
(8.38, 185.10)	Z = −1.77	0.077	20.88
(6.67, 100.91)	Z = −0.38	0.700	
UV-24 h, M (Q1, Q3), (L)	2.00
(1.45, 2.50)	2.05
(1.45, 2.70)	Z = −0.98	0.329	2.05
(1.44, 2.82)	2.20
(1.50, 2.68)	Z = −0.65	0.513	2.05
(1.45, 2.66)	Z = −1.31	0.190	
UPQ-24 h, M (Q1, Q3), (mg/24 h)	138.92
(76.00, 330.54)	149.10
(77.00, 379.76)	Z = −0.07	0.944	119.40
(67.00, 249.86)	142.74
(74.00, 518.99)	Z = −1.79	0.074	137.43
(74.12, 330.60)	Z = −1.65	0.099	
SU, M (Q1, Q3), (mmol/L)	5.60
(4.57, 7.05)	5.60
(4.35, 7.00)	Z = −0.39	0.694	5.38
(4.35, 7.21)	5.44
(4.58, 7.14)	Z = −0.51	0.613	5.58
(4.47, 7.08)	Z = −0.60	0.549	
SC, M (Q1, Q3), (μmol/L)	74.00
(59.00, 95.75)	70.00
(58.00, 90.00)	Z = −1.38	0.169	70.50
(58.88, 85.00)	73.00
(61.50, 97.00)	Z = −1.51	0.131	72.00
(59.00, 93.00)	Z = −0.67	0.506	
SUA, M (Q1, Q3), (μmol/L)	337.50
(279.62, 403.10)	320.10
(261.00, 400.80)	Z = −1.36	0.174	338.50
(282.00, 411.33)	326.00
(263.00, 387.00)	Z = −1.75	0.080	331.00
(273.00, 402.48)	Z = −0.56	0.577	
TCO2, M (Q1, Q3), (mmol/L)	23.60
(21.70, 25.50)	24.10
(22.20, 26.40)	Z = −3.40	<0.001*	23.90
(22.08, 25.13)	24.60
(22.50, 27.55)	Z = −2.67	0.008*	24.00
(22.00, 26.00)	Z = −0.78	0.435	
GFR, M (Q1, Q3),
(mL/min/1.73 m2)	94.92
(71.19, 107.02)	94.81
(74.77, 108.37)	Z = −0.51	0.608	95.93
(75.22, 106.21)	99.05
(75.79, 106.89)	Z = −0.73	0.463	95.09
(73.67, 107.17)	Z = −0.77	0.441	
CysC, M (Q1, Q3), (mg/L)	1.00
(0.79, 1.28)	1.00
(0.82, 1.24)	Z = −0.25	0.803	95.93
(75.22, 106.21)	99.05
(75.79, 106.89)	Z = −0.73	0.463	1.00
(0.81, 1.26)	Z = −0.18	0.855	
β2-MG, M (Q1, Q3), (mg/L)	2.12
(1.68, 3.14)	2.19
(1.74, 3.00)	Z = −0.09	0.932	2.07
(1.71, 2.67)	2.39
(1.79, 3.66)	Z = −2.71	0.007*	2.16
(1.73, 3.10)	Z = −0.35	0.729	
Liver function indicators											
TP, M (Q1, Q3), (g/L)	65.65
(62.02, 69.50)	64.80
(60.70, 69.80)	Z = −1.37	0.172	66.80
(63.27, 71.25)	64.10
(59.30, 68.85)	Z = −3.36	<0.001*	65.40
(61.60, 69.80)	Z = −1.28	0.202	
Alb, M (Q1, Q3), (g/L)	40.30
(37.02, 43.08)	39.80
(36.20, 43.30)	Z = −0.47	0.636	40.80
(38.40, 44.23)	39.30
(36.10, 42.95)	Z = −3.16	0.002*	40.20
(36.90, 43.30)	Z = −1.78	0.076	
PAB, M (Q1, Q3), (mg/L)	113.00
(30.80, 245.00)	334.00
(288.00, 342.00)	Z = −13.52	<0.001*	123.50
(30.80, 278.00)	334.00
(287.00, 342.00)	Z = −6.82	<0.001*	223.00
(35.60, 334.00)	Z = −0.45	0.654	
T-Bil, M (Q1, Q3), (µmol/L)	8.80
(6.40, 12.30)	9.90
(7.10, 13.30)	Z = −2.24	0.025*	8.85
(6.70, 11.90)	9.40
(6.85, 12.45)	Z = −0.78	0.435	9.00
(6.62, 12.70)	Z = −0.38	0.702	
D-Bil, M (Q1, Q3), (µmol/L)	3.10
(2.10, 4.20)	3.07
(2.20, 4.20)	Z = −0.08	0.937	3.10
(2.40, 4.10)	3.00
(2.30, 4.00)	Z = −0.47	0.641	3.10
(2.20, 4.19)	Z = −0.01	0.990	
I-Bil, M (Q1, Q3), (µmol/L)	5.40
(3.70, 8.07)	6.49
(4.30, 9.50)	Z = −3.36	<0.001*	5.70
(3.98, 8.10)	6.00
(3.85, 8.10)	Z = −0.43	0.669	5.70
(3.90, 8.57)	Z = −0.14	0.888	
Blood lipid levels												
TG, M (Q1, Q3), (mmol/L)	1.50
(1.09, 2.32)	1.44
(0.95, 2.25)	Z = −1.70	0.089	1.45
(0.90, 2.41)	1.46
(1.06, 2.35)	Z = −0.60	0.545	1.48
(1.01, 2.33)	Z = −0.43	0.667	
TC, M (Q1, Q3), (mmol/L)	4.33
(3.46, 5.29)	4.45
(3.66, 5.19)	Z = −0.53	0.595	4.37
(3.53, 5.20)	4.28
(3.54, 5.20)	Z = −0.02	0.982	4.38
(3.55, 5.25)	Z = −0.59	0.557	
HDL-C, M (Q1, Q3), (mmol/L)	1.01
(0.81, 1.20)	1.03
(0.84, 1.23)	Z = −1.45	0.146	1.04
(0.87, 1.23)	1.06
(0.88, 1.24)	Z = −0.39	0.698	1.02
(0.84, 1.22)	Z = −1.56	0.119	
LDL-C, M (Q1, Q3), (mmol/L)	2.62
(1.84, 3.39)	2.68
(2.10, 3.37)	Z = −1.58	0.115	2.46
(1.76, 3.18)	2.60
(2.02, 3.58)	Z = −1.62	0.104	2.60
(1.93, 3.38)	Z = −1.39	0.164	
Immune system indicators											
HS-CRP, M (Q1, Q3), (mg/L)	2.21
(0.86, 6.49)	2.47
(0.92, 7.98)	Z = −1.29	0.198	1.98
(0.87, 5.91)	2.07
(0.84, 5.96)	Z = −0.27	0.791	2.18
(0.87, 6.63)	Z = −0.91	0.363	
IgG, M (Q1, Q3), (g/L)	11.97
(9.52, 15.11)	10.81
(8.65, 13.98)	Z = −3.45	<0.001*	11.67
(9.22, 14.36)	10.12
(7.69, 13.22)	Z = −2.91	0.004*	11.59
(9.05, 14.36)	Z = −1.51	0.131	
IgA, M (Q1, Q3), (g/L)	2.85
(2.02, 4.02)	2.50
(1.57, 3.41)	Z = −3.04	0.002*	3.04
(2.09, 4.02)	2.27
(1.43, 3.66)	Z = −2.81	0.005*	2.75
(1.91, 3.85)	Z = −0.62	0.536	
IgM, M (Q1, Q3), (g/L)	0.85
(0.62, 1.23)	0.85
(0.60, 1.17)	Z = −0.86	0.388	0.86
(0.61, 1.30)	0.94
(0.66, 1.34)	Z = −1.08	0.279	0.85
(0.61, 1.28)	Z = −1.27	0.203	
IgE, M (Q1, Q3), (IU/mL)	52.36
(20.10, 313.85)	137.00
(20.10, 638.90)	Z = −3.04	0.002*	31.01
(19.65, 176.00)	228.00
(21.77, 680.00)	Z = −4.88	<0.001*	52.36
(20.10, 455.00)	Z = −0.89	0.373	
C3, M (Q1, Q3), (g/L)	1.09
(0.87, 1.29)	1.11
(0.87, 1.34)	Z = −1.25	0.210	1.04
(0.85, 1.31)	1.09
(0.87, 1.33)	Z = −0.33	0.743	1.09
(0.87, 1.31)	Z = −0.72	0.474	
C4, M (Q1, Q3), (g/L)	0.23
(0.17, 0.36)	0.21
(0.15, 0.29)	Z = −2.65	0.008*	0.23
(0.16, 0.31)	0.22
(0.15, 0.35)	Z = −0.11	0.912	0.22
(0.17, 0.34)	Z = −0.44	0.659	
Urinalysis												
SG, M (Q1, Q3)	1.02
(1.01, 1.03)	1.02
(1.01, 1.02)	Z = −3.20	0.001*	1.02
(1.01, 1.03)	1.02
(1.01, 1.02)	Z = −2.13	0.033*	1.02
(1.01, 1.03)	Z = −0.62	0.538	
pH, M (Q1, Q3)	5.50
(5.50, 6.00)	5.50
(5.50, 6.00)	Z = −1.23	0.220	5.50
(5.00, 6.00)	6.00
(5.50, 6.00)	Z = −1.94	0.053	5.50
(5.50, 6.00)	Z = −0.03	0.976	
Nit, n (%)			χ2 = 0.24	0.625			χ2 = 0.15	0.701		χ2 = 0.15	0.698	
No	417 (95.21)	269 (94.39)			173 (96.11)	101 (94.39)			960 (95.05)			
Yes	21 (4.79)	16 (5.61)			7 (3.89)	6 (5.61)			50 (4.95)			
Pro, n (%)			χ2 = 1.94	0.163			χ2 = 13.17	<0.001*		χ2 = 1.19	0.276	
No	306 (69.86)	185 (64.91)			142 (78.89)	63 (58.88)			696 (68.91)			
Yes	132 (30.14)	100 (35.09)			38 (21.11)	44 (41.12)			314 (31.09)			
Glu, n (%)			χ2 = 2.69	0.101			χ2 = 6.48	0.011*		χ2 = 0.60	0.437	
No	150 (34.25)	81 (28.42)			72 (40.00)	27 (25.23)			330 (32.67)			
Yes	288 (65.75)	204 (71.58)			108 (60.00)	80 (74.77)			680 (67.33)			
Ket, n (%)			χ2 = 3.83	0.050			χ2 = 1.83	0.176		χ2 = 1.27	0.260	
No	388 (88.58)	238 (83.51)			164 (91.11)	92 (85.98)			882 (87.33)			
Yes	50 (11.42)	47 (16.49)			16 (8.89)	15 (14.02)			128 (12.67)			
UBG, n (%)			χ2 = 0.53	0.468			χ2 = 0.15	0.701		χ2 = 1.67	0.197	
No	420 (95.89)	270 (94.74)			176 (97.78)	103 (96.26)			969 (95.94)			
Yes	18 (4.11)	15 (5.26)			4 (2.22)	4 (3.74)			41 (4.06)			
OB, n (%)			χ2 = 0.31	0.581			χ2 = 1.05	0.305		χ2 = 0.37	0.541	
No	354 (80.82)	235 (82.46)			147 (81.67)	82 (76.64)			818 (80.99)			
Yes	84 (19.18)	50 (17.54)			33 (18.33)	25 (23.36)			192 (19.01)			
URBC, M (Q1, Q3), (cells/μL)	0.85
(0.00, 2.80)	1.70
(0.20, 6.00)	Z = −5.48	<0.001*	1.25
(0.00, 4.53)	1.40
(0.15, 8.35)	Z = −1.81	0.070	1.20
(0.00, 4.10)	Z = −1.85	0.064	
UWBC, M (Q1, Q3), (cells/μL)	1.60
(0.00, 9.15)	2.30
(0.30, 8.00)	Z = −2.03	0.042*	1.70
(0.00, 7.88)	2.60
(0.20, 14.72)	Z = −1.33	0.182	2.00
(0.00, 9.40)	Z = −1.07	0.285	
Blood routine												
RBC, M (Q1, Q3), (×10^12/L)	4.54
(4.07, 4.94)	4.52
(4.01, 5.03)	Z = −0.04	0.965	4.67
(4.22, 5.00)	4.61
(4.02, 4.96)	Z = −1.46	0.145	4.58
(4.07, 4.99)	Z = −1.26	0.207	
WBC, M (Q1, Q3), (×10^9/L)	6.73
(5.64, 8.69)	7.04
(5.68, 8.76)	Z = −1.12	0.265	6.76
(5.62, 8.38)	6.92
(5.72, 8.23)	Z = −0.25	0.805	6.84
(5.66, 8.67)	Z = −0.68	0.496	
PLT, M (Q1, Q3), (×10^9/L)	221.50
(181.25, 269.00)	219.00
(175.00, 258.00)	Z = −0.90	0.368	226.00
(175.75, 272.25)	221.00
(174.00, 269.50)	Z = −0.27	0.787	221.00
(177.00, 267.75)	Z = −0.46	0.648	
ANC, M (Q1, Q3), (×10^9/L)	4.13
(3.21, 5.45)	4.20
(3.30, 5.75)	Z = −1.24	0.214	4.03
(3.24, 5.52)	4.20
(3.15, 5.39)	Z = −0.14	0.888	4.14
(3.22, 5.52)	Z = −0.72	0.471	
Neut%, M (Q1, Q3), (%)	61.40
(53.82, 67.10)	60.30
(53.60, 68.50)	Z = −0.64	0.520	60.20
(51.30, 68.03)	59.70
(54.20, 68.80)	Z = −0.60	0.546	60.75
(53.60, 67.80)	Z = −0.69	0.488	
ALC, M (Q1, Q3), (×10^9/L)	1.86
(1.40, 2.38)	1.88
(1.43, 2.48)	Z = −1.02	0.308	1.90
(1.36, 2.43)	1.77
(1.39, 2.23)	Z = −0.89	0.372	1.86
(1.40, 2.42)	Z = −0.42	0.677	
Lymph%, M (Q1, Q3), (%)	28.30
(21.72, 34.20)	27.80
(20.10, 34.90)	Z = −0.60	0.548	28.50
(21.67, 36.25)	28.10
(21.45, 32.60)	Z = −1.29	0.196	28.20
(21.22, 34.60)	Z = −0.70	0.481	
Mono%, M (Q1, Q3), (%)	7.60
(6.30, 9.00)	7.10
(5.80, 8.60)	Z = −3.06	0.002*	7.40
(6.30, 9.12)	7.20
(6.00, 8.50)	Z = −0.38	0.702	7.40
(6.00, 8.90)	Z = −0.17	0.868	
AMC, M (Q1, Q3), (×10^9/L)	0.51
(0.41,0.64)	0.48
(0.40, 0.63)	Z = −1.20	0.229	0.49
(0.38, 0.65)	0.50
(0.39, 0.59)	Z = −0.38	0.702	0.50
(0.40, 0.63)	Z = −0.95	0.342	
ESR, M (Q1, Q3), (mm/h)	29.00
(15.00, 57.00)	25.00
(11.00, 51.00)	Z = −1.71	0.088	30.00
(12.78, 50.25)	20.00
(9.50, 55.00)	Z = −1.18	0.239	27.00
(12.00, 55.00)	Z = −1.16	0.248	
Notes:

Categorical data are shown as frequencies (%) and continuous data as median (quartile); the classification of UA/CR is as follows: normal value is less than 30 mg/g, microalbuminuria ranges from 30 to 300 mg/g, and macroalbuminuria is greater than 300 mg/g; Z: Mann-Whitney test; χ2: Chi-square test. The asterisk (*) and bold font indicate that the result is statistically significant (P < 0.05).

UA/CR, Urine Microalbumin/Creatinine Ratio; UMA-24h, Urinary Microalbumin 24-hour; UV-24h, Urine Volume 24-hour; UPQ-24h, Urine Protein Quantification 24-hour; SU, Serum Urea; SC, Serum Creatinine; SUA, Serum Uric Acid; TCO2, Total Carbon Dioxide; GFR, Glomerular Filtration Rate; CysC, Cystatin C; β2-MG, Beta-2 Microglobulin; TP, Total Protein; Alb, Albumin; PAB, Prealbumin; T-Bil, Total Bilirubin; D-Bil, Direct Bilirubin; I-Bil, Indirect Bilirubin; TG, Triglycerides; TC, Total Cholesterol; HDL-C, High-Density Lipoprotein Cholesterol; LDL-C, Low-Density Lipoprotein Cholesterol; HS-CRP, High-Sensitivity C-Reactive Protein; C3, Complement Component 3; C4, Complement Component 4; SG, Specific Gravity; Nit, Nitrite; Pro, Protein; Glu, Glucose; Ket, Ketones; UBG, Urobilinogen; OB, Occult Blood; URBC, Urine Red Blood Cells; UWBC, Urine White Blood Cells; RBC, Red Blood Cells; WBC, White Blood Cells; PLT, Platelets; ANC, Absolute Neutrophil Count; Neut%, Neutrophil Percentage; ALC, Absolute Lymphocyte Count; Lymph%, Lymphocyte Percentage; Mono%, Monocyte Percentage; AMC, Absolute Monocyte Count; ESR, Erythrocyte Sedimentation Rate.

Development of the predictive model

In a multivariate logistic regression analysis, HbA1c, PCP-2h, DPN, TCO2, PAB, T-Bil, I-Bil, IgE, URBC, UI and UR were identified as independent predictive factors for DBD, and a nomogram was constructed. The method for using the nomogram is provided in Appendix 1. Detailed results can be found in Table 4 and Fig. 2. Notably, T-Bil emerged as a protective factor against DBD.

Table 4 Multivariate logistic regression analysis of DBD.

Variables	β	OR (95% CI)	Z value	P-value	
(Intercept)	−8.385	0.000 [0.000–0.001]	−9.485	<0.001*	
HbA1c	0.097	1.101 [1.012–1.200]	2.232	0.026*	
PCP-2h	0.089	1.093 [1.005–1.188]	2.074	0.038*	
DPN	1.845	6.326 [3.861–10.651]	7.14	<0.001*	
UMA-24h	0.000	1.093 [1.005–1.188]	1.485	0.137	
TCO2	0.031	1.031 [1.013–1.070]	2.506	0.012*	
PAB	0.011	1.012 [1.009–1.014]	11.29	<0.001*	
T-Bil	−0.082	0.921 [0.844–0.985]	−2.167	0.03*	
I-Bil	0.196	1.217 [1.100–1.376]	3.52	<0.001*	
IgA	0.119	1.127 [0.971–1.310]	1.563	0.118	
IgE	0.002	1.002 [1.001–1.002]	5.411	<0.001*	
URBC	0.004	1.004 [1.001–1.008]	2.322	0.02*	
UI	0.714	2.042 [1.330–3.157]	3.243	0.001*	
UR	1.510	4.529 [2.869–7.263]	6.388	<0.001*	
Notes:

β is the regression coefficient.

OR, odds ratio; CI, confidence interval; HbA1c, Glycated Hemoglobin; DPN, Diabetic Peripheral Neuropathy; PCP-2h, Postprandial C-peptide 2-hour; UMA-24h, Urinary Microalbumin 24-hour; TCO2, Total Carbon Dioxide; PAB, Prealbumin; T-Bil, Total Bilirubin; I-Bil, Indirect Bilirubin; URBC, Urinary Red Blood Cells; UI, Urinary Incontinence; UR, Urinary Retention; the asterisk (*) indicates that the result is statistically significant (P < 0.05).

Figure 2 Nomogram for DBD in type 2 diabetes patients.

Note: A nomogram for DBD was established in this study and correlated with HbA1c, PCP-2h, DPN, TCO2, PAB, T-Bil, I-Bil, IgE, URBC, UI and UR, the points of the 11 variables identified on the scale are summed to obtain the total number of points. A vertical line is then drawn from the total points scale to the last axis to obtain the corresponding probability of DBD. Abbreviations: HbA1c, Glycated Hemoglobin; PCP-2h, Postprandial C-peptide 2-hour; DPN, Diabetic Peripheral Neuropathy; TCO2, Total Carbon Dioxide; PAB, Prealbumin; T-Bil, Total Bilirubin; I-Bil, Indirect Bilirubin; URBC, Urinary Red Blood Cells; UI, Urinary Incontinence; UR, Urinary Retention.

Model performance

The discriminative ability of the predictive model was assessed using the ROC curve, which was analyzed for both the complete model and the top four individual factors ranked by AUC value. Detailed results are presented in Table 5, and the ROC curves are depicted in Fig. 3. The analysis yielded an AUC value of 0.897 (95% CI [0.874–0.920]) with an optimal cutoff point of 0.415, demonstrating a sensitivity of 0.832 and a specificity of 0.826. These findings underscore the model’s robust predictive ability for identifying the occurrence of DBD. Additionally, the Hosmer-Lemeshow test (P = 0.319) indicated a good model fit. The calibration curve (P = 0.782) showed a close alignment between the actual and predicted values, demonstrating that the model’s predictions of DBD risk in T2DM patients closely correspond to the observed risk, as illustrated in Fig. 4.

Table 5 ROC curve analysis of factors and the predictive model with DBD.

Variable(s)	AUC	Specificity	Sensitivity	95% CI	
Model	0.897	0.826	0.832	[0.874–0.920]	
PAB	0.797	0.804	0.811	[0.761–0.832]	
UR	0.658	0.797	0.519	[0.623–0.693]	
DPN	0.640	0.445	0.835	[0.608–0.672]	
URBC	0.618	0.409	0.786	[0.561–0.643]	
TCO2	0.575	0.820	0.298	[0.532–0.618]	
I-Bil	0.574	0.584	0.565	[0.531–0.617]	
HbA1c	0.569	0.815	0.326	[0.526–0.613]	
IgE	0.567	0.639	0.516	[0.523–0.611]	
PCP-2h	0.562	0.728	0.389	[0.519–0.606]	
UI	0.566	0.669	0.463	[0.530–0.602]	
T-Bil	0.549	0.600	0.505	[0.507–0.592]	
Note:

AUC, area under the curve; CI, confidence interval; PAB, Prealbumin; UR, Urinary Retention; DPN, Diabetic Peripheral Neuropathy; URBC, Urinary Red Blood Cells; TCO2, Total Carbon Dioxide; I-Bil, Indirect Bilirubin; HbA1c, Glycated Hemoglobin; PCP-2h, Postprandial C-peptide 2-hour; UI, Urinary Incontinence; T-Bil, Total Bilirubin.

Figure 3 ROC curve of the predictive model for DBD and the top four individual factors ranked by AUC value.

Abbreviations: AUC, area under the curve; CI, confidence interval; PAB, Prealbumin; UR, Urinary retention; DPN, Diabetic peripheral neuropathy; UWBC, Urine white blood cells.

Figure 4 Calibration curve of DBD predictive nomogram.

Note: The gray line represents the perfect prediction of the ideal model; the dashed line indicates the performance of the model; the solid line represents the revised estimation.

Validation of the predictive model

In this study, we validated the model using data from both the training and validation sets. In the training set, The predictive model underwent internally validated by Bootstrap resampling 1,000 times, ensuring its stability (Accuracy = 0.810, Kappa = 0.600). The AUC remained at 0.896 (95% CI [0.873–0.919]). In the validation set, the AUC value was 0.862 (95% CI [0.816–0.908]), with an optimal cutoff of 0.493, a sensitivity of 0.710, and a specificity of 0.894. These findings suggest that the nomogram model effectively distinguishes between the risk of DBD and Non-DBD occurrence during early screening.

Calibration curves were used to assess the degree of model fit and prediction accuracy. The calibration curve represents a scatter plot of the actual probability of occurrence vs. the predicted probability (Chen et al., 2024). The closer the calibration curve is to a fitted straight line, the better the expected value matches the measured value, indicating greater accuracy of the model (Chen et al., 2024). The results show that the calibration curves fit well with the actual curves, suggesting that the predicted probability of DBD aligns with the exact probability for both data sets (see Figs. 5A, 5B).

Figure 5 Calibration curve for the internal validation of the predictive nomogram.

Note: (A) represents the calibration curve of the training set; (B) represents the calibration curve of the validation set; “Ideal” refers to the ideal line, representing the reference line for the perfect model; “Apparent” is the apparent line directly calculated from the sample; “Bias-corrected” is the bias-corrected line adjusted through 1,000 bootstrap resamplings.

The clinical utility of the nomogram

Figures 6A, 6B illustrate the clinical utility and net benefit of the DBD predictive model across different high-risk thresholds. In the training set (Fig. 6A), the decision curve analysis (DCA) demonstrates that the model provides a net benefit within a threshold probability range of 3% to 73%, indicating its strong predictive performance over a wide range of clinical scenarios. In the validation set (Fig. 6B), the DCA results reveal that the model offers a net benefit within a threshold probability range of 4% to 71%. This range is slightly narrower than that observed in the training set, reflecting the model’s consistent yet slightly reduced performance in the external validation cohort. Overall, the DCA results highlight the robust clinical applicability of the DBD predictive model in identifying high-risk patients, with meaningful net benefits observed in both the training and validation sets.

Figure 6 The DCA of the predictive nomogram.

Note: (A) represents the DCA of the training set; (B) represents the DCA of the validation set.DCA shows the clinical usefulness of the predictive nomogram. The solid gray line indicates that all patients developed DBD. The solid black line indicates that no patient developed DBD. The dark red curve indicates the performance of the nomogram, and the light red curve represents the 95% confidence interval of the DCA. Abbreviation: DCA, Decision curve analysis.

Figures 7A, 7B provides a comprehensive assessment of the clinical efficacy of the DBD risk prediction nomogram across both training and validation sets. In the training set (Fig. 7A), when the threshold probability exceeds 79%, the subgroup identified as high-risk DBD by the nomogram closely corresponds to the actual high-risk DBD population. This alignment demonstrates the nomogram’s strong predictive accuracy and clinical relevance in this setting.

Figure 7 The CIC of the predictive nomogram.

Note: (A) represents the CIC of the training set; (B) represents the CIC of the validation set. CIC demonstrates the clinical efficiency of the predictive nomogram. The blue curve indicates the number of persons who are classified as positive (high risk) by the prediction nomogram at each threshold probability. The red curve depicts the number of true positives at each threshold probability. Abbreviation: CIC, Clinical impact curve.

Similarly, in the validation set (Fig. 7B), the nomogram exhibits robust performance. Threshold probabilities exceeding 80% show a strong association between the predicted high-risk group and the actual high-risk DBD population.

Discussion

In this study, a risk prediction model for DBD was developed and internally validated as well as externally validated. Through multivariate analysis, several key variables were included in the final model: HbA1c, PCP-2h, DPN, TCO2, PAB, T-Bil, I-Bil, IgE, URBC, UI and UR. The performance and clinical relevance of the prediction model were comprehensively evaluated using various analytical methods. The results demonstrated that the model exhibits good efficacy and clinical applicability in predicting DBD.

Surprisingly, this study found that 78.25% of patients with DBD did not have DN, and the occurrence of DBD was not significantly associated with the presence of DN (P = 0.789). This suggests that DBD may be an independent complication of diabetes rather than an adjunct manifestation of DN, revealing the multifaceted impact of diabetes on the urinary system. Previous studies have indicated that DN typically results from diabetes-induced damage to the renal microvasculature and renal tubules (Yu & Bonventre, 2018), whereas DBD is likely primarily caused by autonomic neuropathy. Diabetic neuropathy, even without renal impairment, can directly affect bladder nerve control, leading to bladder dysfunction (Daneshgari et al., 2009). This finding presents new challenges for clinical management: traditionally, DN and DBD have been considered linked complications, with bladder dysfunction often accompanying renal impairment. However, this study demonstrates that DBD can occur independently of DN, necessitating the inclusion of DBD screening and early intervention in the management of diabetes patients, even in the absence of DN symptoms.

The results of this study indicate that HbA1c (OR: 1.101, 95% CI [1.012–1.200]) and DPN (OR: 6.326, 95% CI [3.861–10.651]) play crucial roles in predicting DBD. HbA1c, an important indicator of long-term glycemic control, reflects chronic hyperglycemia in T2DM patients. Elevated HbA1c levels promote the accumulation of advanced glycation end products, which impair the structure and function of microvasculature, leading to microvascular complications (Khalid, Petroianu & Adem, 2022). Studies have shown that microvascular complications not only affect the blood supply and oxygenation of nerves but ultimately result in nerve fiber degeneration and apoptosis, triggering DPN (Tesfaye & Selvarajah, 2012). Notably, DPN extends beyond peripheral nervous system damage to affect the autonomic nervous system, damaging sympathetic and parasympathetic nerves. This directly impairs bladder sensory and motor nerve functions, leading to reduced bladder sensation and disrupted voiding reflexes (Feldman et al., 2019). Therefore, effective glycemic control and early intervention for neuropathy are paramount.

Univariate analysis in this study revealed that PCP-2h levels in DBD patients were significantly lower than those in Non-DBD patients, similar to previous research findings (Saisho, 2016). Elevated PCP-2h levels indicate good responsiveness of pancreatic β-cells, helping to reduce postprandial blood glucose levels and potentially lowering the risk of DBD (Tai et al., 2016; Saisho, 2016). However, surprisingly, multivariate analysis in our study identified PCP-2h levels (OR: 1.093, 95% CI [1.005–1.188]) as a risk factor for the occurrence of DBD. To gain deeper insight into this phenomenon, a comparative analysis was conducted between significant variables from the univariate analysis and PCP-2h levels. The results revealed a significant confounding effect of HbA1c on PCP-2h levels. This finding indicates that while PCP-2h levels can reflect pancreatic β-cell functionality to some extent (Saisho, 2016), their relationship with DBD is neither direct nor singular but rather influenced by the complex state of glycemic control in patients. This suggests a complex interaction between PCP-2h levels, HbA1c, and the development of DBD, which necessitates further investigation in future studies.

This study revealed a significant finding: higher TCO2 levels are significantly associated with an increased risk of DBD (OR: 1.031, 95% CI [1.013–1.070]). This suggests that acid-base balance may play a crucial role in the pathogenesis of DBD. In patients with T2DM, persistent hyperglycemia due to insulin resistance and insufficient insulin secretion can lead to metabolic acidosis (Farwell & Taylor, 2008). This condition may increase the production of free radicals, resulting in oxidative stress. Oxidative stress not only directly damages cell membranes, DNA, and intracellular proteins (Powell & Gehring, 2023), but it may also harm nerve cells and bladder wall cells, thereby impairing the normal functions of nerves and the bladder (Xu et al., 2022). This finding provides new perspectives and potential intervention targets for the prevention and treatment of DBD.

Our results emphasize the importance of liver function indicators in assessing the risk of DBD. The findings show that elevated levels of PAB (OR = 1.012, 95% CI [1.009–1.014]) and I-Bil (OR = 1.217, 95% CI [1.100–1.376]) are associated with the occurrence of DBD. This may suggest a potential role of liver dysfunction in the pathogenesis of DBD, possibly leading to metabolic disorders, increased oxidative stress, and enhanced inflammatory responses (Michael et al., 2000; Szasz et al., 2016). These factors collectively exert adverse effects on cells and tissues (Powell & Gehring, 2023), particularly affecting the normal functions of the nervous and urinary systems. Additionally, an interesting observation was made in this study: elevated T-Bil levels (OR = 0.921, 95% CI [0.844–0.985]) are associated with a reduced risk of DBD. This finding contradicts the notion that I-Bil is a risk factor for DBD but may reflect the complex physiological roles of bilirubin. The observed phenomenon may be attributed to the natural antioxidant properties of bilirubin, which help alleviate diabetes-related oxidative stress (Vítek, 2012), potentially reducing the risk of microvascular damage and neuropathy (Tesfaye & Selvarajah, 2012; Khalid, Petroianu & Adem, 2022).

It is noteworthy that elevated IgE levels play a potential role in predicting DBD (OR = 1.002, 95% CI [1.001–1.002]). The increase in IgE may reflect immune system activation and disruption of mucosal barriers under prolonged hyperglycemic conditions, a phenomenon particularly common in patients with T2DM. This immune response could facilitate pathogen invasion, triggering localized immune reactions that cause damage to bladder tissues (Velloso, Eizirik & Cnop, 2013; Khalid, Petroianu & Adem, 2022). Moreover, elevated IgE levels may exacerbate bladder dysfunction by promoting inflammation, which contributes to increased damage and fibrosis in the bladder smooth muscle (Wang et al., 2016; Borsodi et al., 2023). These findings provide a new perspective, suggesting that immune regulation may play a more significant role in the pathogenesis of DBD, warranting further exploration in future research.

This study is the first to reveal that elevated URBC count (OR = 1.004, 95% CI [1.001–1.008]) is significantly associated with an increased risk of DBD in patients with T2DM. High URBC counts reflect microvascular damage or an inflammatory state within the urinary system. In T2DM patients, chronic hyperglycemia is a key factor leading to endothelial damage and microvascular complications (Stehouwer, 2018; Khalid, Petroianu & Adem, 2022). Such microvascular damage can result in inadequate blood supply to the bladder wall, impairing bladder contraction and voiding capability (Wang et al., 2023). Furthermore, a high-glucose environment promotes the increase of reactive oxygen species, triggering oxidative stress and cellular damage (Khalid, Petroianu & Adem, 2022). This process can induce inflammatory responses, exacerbating urinary tract irritation and contributing to further urinary system damage. These combined factors may ultimately lead to the development of DBD.

This study highlights the critical role of UI (OR: 2.042, 95% CI [1.330–3.157]) and UR (OR: 4.529, 95% CI [2.869–7.263]) in DBD. UI, a common symptom of DBD, aligns with previous studies (Daneshgari et al., 2009), suggesting that diabetes-induced neuropathy may impair bladder storage function, thus leading to UI. Furthermore, the occurrence of UR is closely linked to diabetic neuropathy, particularly in cases involving autonomic nerve damage, which impairs bladder emptying and exacerbates bladder dysfunction (Feldman et al., 2019). Our findings confirm UI and UR as independent risk factors for DBD, indicating multiple dysfunctions in bladder storage and emptying in diabetic patients. These findings not only deepen our understanding of the mechanisms underlying bladder symptoms in diabetes, but also provide crucial theoretical grounds for clinical interventions and patient management, emphasizing the importance of early bladder function assessment and intervention in diabetic patients.

The strength of this study lies in the use of a large-scale retrospective data system to analyze risk factors associated with DBD in patients with T2DM, and the successful development of a predictive model. First, the model integrates not only traditional diabetes-related indicators but also multidimensional data such as liver and kidney function markers and immune biomarkers, thereby improving predictive accuracy and clinical applicability. Second, the model demonstrated strong predictive power and calibration through both internal and external validation, Further supporting its potential clinical utility. Moreover, this study not only analyzed traditional hyperglycemia markers but also explored the potential role of novel biomarkers in the pathogenesis of DBD, providing new directions and intervention targets for future research. Finally, the study incorporated the temporal sequence between T2DM and the onset of DBD, enhancing the interpretability of the results.

We also acknowledge the limitations of this study. First, as a retrospective study, it relies on the detailed records of medical case files, with the presence of predictive factors and prognostic outcomes dependent on these historical data. However, this reliance may introduce inherent information bias. Second, convenience sampling was used to extract data from the electronic medical records of two hospitals. Although this method allows for rapid sample collection, it may introduce selection bias and fail to comprehensively represent the characteristics of all T2DM patients, potentially leading to the underestimation or overestimation of certain groups. Additionally, the study data were sourced from only two general tertiary hospitals in Shenzhen. While these hospitals have a diverse patient population, the geographic limitation to Shenzhen restricts the generalizability of the results. To improve the external validity of the findings, future studies should adopt more stringent random sampling methods, such as stratified random sampling, and expand the data source to include patients from different regions and healthcare levels, covering a broader spectrum of T2DM patients. Furthermore, future research could consider a multi-center, prospective design to further explore causal relationships, analyze potential influencing factors, and validate the generalizability and stability of the model across different patient populations, thereby providing more reliable scientific evidence.

Conclusion

This study developed and both internally and externally validated a risk prediction model for DBD in patients with T2DM, and constructed a nomogram. The model demonstrated excellent discrimination, calibration, and clinical validity, providing healthcare professionals with an effective tool to reduce the incidence of DBD and improve patients’ quality of life.

Supplemental Information

Supplemental Information 1 287 samples in the validation set.

Supplemental Information 2 723 samples in the training set.

Supplemental Information 3 Assigned values of categorical variables.

Supplemental Information 4 STROBE Statement.

Supplemental Information 5 Instructions for using the nomogram: A tool for predicting diabetic bladder dysfunction (DBD) in type 2 diabetes patients.

Additional Information and Declarations

Competing Interests

Author Contributions

Human Ethics

Data Availability

The authors declare that they have no competing interests.

Yingjie Hu analyzed the data, authored or reviewed drafts of the article, and approved the final draft.

Fengming Hao analyzed the data, authored or reviewed drafts of the article, and approved the final draft.

Ying Wang performed the experiments, prepared figures and/or tables, and approved the final draft.

Ling Chen conceived and designed the experiments, authored or reviewed drafts of the article, and approved the final draft.

Lihua Wen performed the experiments, prepared figures and/or tables, and approved the final draft.

Jue Li performed the experiments, prepared figures and/or tables, and approved the final draft.

Wei Ren conceived and designed the experiments, authored or reviewed drafts of the article, and approved the final draft.

Wenzhi Cai conceived and designed the experiments, authored or reviewed drafts of the article, and approved the final draft.

The following information was supplied relating to ethical approvals (i.e., approving body and any reference numbers):

The Ethics Committee of Shenzhen Hospital of Southern Medical University has granted ethical approval for the study to be conducted within its facilities, under the Ethical Application Reference: NYSZYYEC20230086.

The following information was supplied regarding data availability:

The raw data is available in the Supplemental Files.

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
