# Peer review of "A nomogram for predicting bladder dysfunction in patients with type 2 diabetes mellitus: a retrospective study"

_PeerJ, doi:10.7717/peerj.18872_

## Round 0.1 · original submission · Major Revisions

To improve this work, please address the concerns raised by both reviewers.

Reviewer 1 ·

Basic reporting

no comment

Experimental design

Why does *** appear instead of the ethics committee approval number on line 164?
Why was *** used on line 65?

Validity of the findings

no comment

Additional comments

The work presented by Hu and colleagues demonstrates a tool for predicting diabetic bladder dysfunction (DBD), which is of great importance in the clinical context, considering that diagnosing DBD is not straightforward. This is a novel approach with potential, but the following considerations may contribute to improving the text.

Please check line 32: is the year 2023 in the phrase the reference year of the citation?
Please check the spacing on lines 36 and 93.
On lines 69-71, could you separate the inclusion criteria into one paragraph and the exclusion criteria into another?
In this study, HbA1c was considered statistically significant for predicting DBD. However, in both groups, with or without DBD, it is elevated. How could this tool help in diagnosis if HbA1c is elevated in both groups?
Regarding the tables, Table 1 would be clearer if organized as follows: Table 1 - Demographic Information; Table 2 - Fundamental Information about T2DM; Table 3 - Laboratory Exams.
If we consider sex as a dependent variable, would the nomogram be presented in the same way?
What symptoms did patients with DBD present? Why didn’t you consider bladder symptoms as a variable to compose the nomogram?
Were DM complications compared between individuals with and without DBD? If sex was considered, would the results be the same among those with and without DBD?
In the discussion, could you address more the importance of patients manifesting DBD without having diabetic nephropathy? This can help in understanding these two dysfunctions, where the patient presents bladder dysfunction without diabetic nephropathy.
I believe that, as this is a tool for clinical use, a better explanation of the use of the nomogram could be added to the article, perhaps in supplementary material, as this would improve the overall understanding of how to apply it in practice.

Reviewer 2 ·

Basic reporting

no comment

Experimental design

The current study used a retrospective design but did not elaborate on whether the sources of data and the collection process were representative. It is recommended that the authors further elucidate the randomization and rationale for sample selection to ensure that the experimental design accurately reflects the incidence of DBD in a broad population of patients with T2DM and to enhance the extrapolation of the findings.

Validity of the findings

Although the model demonstrated good calibration and discrimination in internal validation (AUC=0.875), the current study lacks external validation. It is recommended that the authors conduct a further external dataset validation to ensure the applicability of the model in other diverse patient populations and to enhance the broad applicability and reliability of the results.

Additional comments

1. The wording /English language needs improving throughout.
2. Please better discuss the strengths and limitations of the study.

---

## Round 0.2 · accepted · Accept

The authors have extensively revised their work according to the reviewers'suggestions, so that the paper is now acceptable for publication.

Reviewer 2 ·

Basic reporting

no comment

Experimental design

no comment

Validity of the findings

no comment

Additional comments

The authors addressed my concerns.